# The Impact of Sensory Perceptions and Country-of-Origin Practices on Consumer Preferences for Rice: A Comparative Study of China and Thailand

**DOI:** 10.3390/foods14040603

**Published:** 2025-02-12

**Authors:** Tanapon Srisukwatanachai, Baichen Jiang, Achara Boonkong, Fallah Samuel Kassoh, Sutthawongwadee Senawin

**Affiliations:** 1College of Economics and Management, South China Agricultural University, Guangzhou 510642, China; tanapon.th@gmail.com (T.S.); acharabk@hotmail.com (A.B.); fskassoh@yahoo.com (F.S.K.); 2Sierra Leone Agricultural Research Institute, P. M. B. 1313, Tower Hill, Freetown 47235, Sierra Leone; 3Faculty of Agriculture, Kasetsart University, Bangkok 10900, Thailand; sutthawongwadee.se@ku.th

**Keywords:** sensory perception, consumer behavior, fragrant rice, willingness to pay, choice experiment

## Abstract

This investigation scrutinizes the impact of sensory perceptions and country of origin (COO) on consumer inclinations for aromatic rice in China and Thailand, elucidating pivotal sensory characteristics and cross-cultural variances in purchasing behavior. A choice experiment (CE) involving 1330 participants from Guangzhou and Bangkok assessed attributes such as fragrance, grain quality, certification, and pricing. Structural Equation Modeling (SEM) was used to examine correlations between sensory perceptions, COO, and willingness to pay (WTP). The findings indicate that fragrance and grain integrity substantially influence preferences, with Chinese consumers more inclined to buy premium-certified rice due to escalating incomes and food safety apprehensions, whereas Thai consumers emphasize domestically produced rice and demonstrate pronounced ethnocentrism. Price sensitivity and brand allegiance similarly affect both markets. This study underscores the significance of enhancing sensory and COO attributes to bolster the global competitiveness of aromatic rice, and it provides pragmatic insights for quality assurance, certification, and culturally nuanced marketing strategies.

## 1. Introduction

Rice constitutes a fundamental element of global alimentary security, functioning as a dietary staple for over half of the world’s populace. Its importance is particularly accentuated in Asia, where it serves as a nutritional mainstay. The international rice market, appraised at USD 293.8 billion in 2021, is anticipated to expand to USD 356 billion by 2030, indicating a consistent Compound Annual Growth Rate (CAGR) of 2.6% [1]. This expansion emphasizes the essential role of rice in worldwide food systems, with per capita annual consumption averaging 67.5 kg in 2022. Prominent industry stakeholders, such as MGP Components and Rice Bran Technologies, are leveraging this demand to innovate and develop value-enhanced rice products. Nevertheless, despite vigorous production levels, challenges remain. The 2022/23 agricultural year witnessed global rice production estimated at 503.3 million tons; however, an 8.7-million-tonne shortfall was projected due to escalating demand and production hindrances in major rice-producing countries like China and Pakistan. Geopolitical disruptions, exemplified by the Russia–Ukraine conflict, have further strained rice supplies, establishing rice as a vital alternative to wheat and corn. This discrepancy is particularly pronounced in economically precarious regions such as Bangladesh and Vietnam, underscoring the pressing necessity for strategies that address supply chain resilience and alimentary security [2].

Sensory characteristics—flavor, fragrance, texture, and appearance—play a crucial role in molding consumer inclinations, particularly for premium rice cultivars such as Jasmine, Basmati, and Phka Rumduol. These attributes elevate aromatic rice beyond its function as a dietary staple to a cultural and economic asset for producing nations [3]. Sensory excellence not only amplifies consumer gratification but also contributes significantly to market value, reinforcing competitiveness in the global marketplace. Platforms such as the World’s Best Rice Award, established in 2009 by The Rice Trader (TRT), illustrate the emphasis on sensory quality. This competition meticulously evaluates entries based on flavor, fragrance, and texture, enabling nations like Thailand, Vietnam, and the U.S. to present their finest aromatic rice varieties. Such platforms bolster the economic and cultural significance of sensory characteristics in cultivating global consumer loyalty and enhancing market reputation [4]. Consequently, sensory perceptions are posited to exert a considerable influence on consumer inclinations, particularly through their effects on purchasing decisions and willingness to pay (WTP). Specifically, fragrance and texture are anticipated to positively affect WTP, rendering them critical determinants for producers striving to optimize product value.

Country of origin (COO) also plays a pivotal role in molding consumer behavior by signaling quality, safety, and authenticity [5,6,7]. Thai Jasmine rice, renowned for its unparalleled fragrance and grain quality, establishes a global benchmark for premium rice, while Vietnamese and Cambodian rice are accruing acknowledgment for competitive pricing and certifications. Chinese aromatic rice, such as black rice, appeals to health-conscious consumers due to its nutritional advantages. Empirical investigations suggest that COO influences consumer preferences by augmenting perceptions of quality and trust, particularly in markets where food safety apprehensions are prevalent. For instance, Chinese consumers increasingly favor branded and certified rice, propelled by escalating awareness of food safety and rising disposable incomes [8,9,10]. Conversely, Thai consumers exhibit pronounced ethnocentric inclinations, favoring domestically produced rice that aligns with cultural pride and sensory expectations, such as fragrance and texture [11,12]. Given these patterns, it is hypothesized that Chinese consumers prioritize certification and branding, while Thai consumers demonstrate a stronger preference for locally produced rice, reflecting cultural pride and confidence in domestic production.

The interaction between sensory perceptions and COO is further influenced by socio-economic and cultural dynamics. Chinese consumers, with increasing incomes and heightened health awareness, manifest a preference for rice that accentuates certification and branding, valuing quality, and safety as principal purchasing criteria [13]. Meanwhile, Thai consumers, entrenched in strong ethnocentric tendencies, prefer rice that embodies sensory superiority, such as fragrance and texture, over certifications. This divergence underscores the necessity for customized marketing strategies that align with the distinctive priorities of each market segment. Certification systems, such as Thailand’s Geographical Indications (GIs), enhance consumer trust and product authenticity, while analogous systems in Vietnam and Cambodia reinforce the competitiveness of their fragrant rice varieties. These factors not only affect WTP but also elucidate the interplay of sensory, economic, and cultural factors propelling rice consumption behavior.

This study addresses these dynamics by investigating how sensory perceptions and COO shape consumer preferences for fragrant rice in China and Thailand. This research also examines cross-cultural discrepancies in consumer behavior, with a particular emphasis on how sensory and COO attributes influence purchasing decisions, certification preferences, and ethnocentrism. These findings aim to provide actionable insights for policymakers, producers, and marketers, enabling them to align production and marketing strategies with evolving consumer priorities. By leveraging these insights, stakeholders can enhance the global competitiveness of fragrant rice, addressing cultural and sensory priorities such as certification in China and sensory excellence in Thailand.

## 2. Literature Review

The preferences of consumers and the sensory attributes associated with rice among Thai and Chinese clientele are subject to a multitude of influences, encompassing quality certifications, sensory characteristics, country of origin, and socio-economic determinants. Each demographic displays unique preferences that are sculpted by an interplay of cultural, economic, and sensory factors. The subsequent sections will examine these preferences and attributes in detail, emphasizing significant findings derived from the research.

### 2.1. Quality and Certification Preferences

Green and organic certifications: Both Chinese and Thai consumers exhibit a marked inclination towards rice that bears green and organic certifications. Nonetheless, Chinese consumers manifest a more pronounced preference for these certifications relative to their Thai counterparts, likely attributable to an elevated awareness of health issues and income levels prevalent in China [13].

Traceability and brand labels: Chinese consumers exhibit a greater willingness to incur additional costs for traceability labels, indicative of their apprehensions regarding food safety and quality. Conversely, Thai consumers assign more substantial value to brand labels, which may stem from a heightened awareness of brands and trust within local markets [13].

### 2.2. Sensory Attributes

Taste and aroma: The quality of taste constitutes a paramount consideration for Chinese consumers, who demonstrate a considerable willingness to pay for enhanced taste attributes. Conversely, aroma, particularly the fragrance associated with Jasmine rice, emerges as a pivotal factor for Thai consumers, who tend to prioritize visual appeal and cooking characteristics over taste [12,14].

Texture and color: The texture of rice, encompassing attributes such as hardness and chewiness, serves as a vital determinant of consumer preference across both demographics. White rice is generally favored over brown rice due to its superior texture and color, although nutritional information has the potential to positively influence consumer perceptions [15,16].

### 2.3. Socio-Economic Influences

Income levels: The preferences exhibited by consumers for rice attributes demonstrate significant variation in accordance with income levels. Higher-income demographics typically prioritize taste and quality, whereas lower-income groups tend to emphasize resilience and fundamental quality attributes [17].

Health and environmental concerns: there exists a burgeoning trend among Thai consumers to contemplate healthfulness and reducing carbon footprints when procuring rice, signifying an augmented consciousness regarding the environmental and health ramifications of their choices [18].

### 2.4. Cultural and Regional Differences

Cultural preferences: The phenomenon termed “Jasminization” in Southeast Asia, wherein consumers adopt Thai inclinations for rice fragrance and texture, underscores the cultural determinants influencing rice preferences. This trend is particularly pronounced in nations such as the Philippines and Cambodia [12]. Gender and decision making: women, particularly those who possess empowerment in grocery decision-making roles, significantly contribute to the demand for aromatic rice, suggesting that gender dynamics can substantially shape consumer preferences [12].

### 2.5. Country of Origin (COO)

The implications of COO labeling can profoundly affect consumer perceptions and their willingness to pay (WTP). For example, although COO labels tend to enhance consumer attitudes, they may conversely detract from WTP, functioning as a latent mediator in the relationship between attitudes and WTP [19].

Within the framework of consumer behavior among Chinese and Thai populations, COO labels represent an integral component of a more extensive array of quality indicators, which encompass brand identity, traceability, and certifications pertaining to sustainability and organic practices. Thai consumers exhibit an elevated WTP for brand identifiers and certifications, attributable to heightened awareness and trust, whereas Chinese consumers prioritize traceability labels, motivated by increasing income levels and a growing consciousness regarding health [13].

COO labeling can function as a strategic competitive advantage within the marketplace, as evidenced by historical instances where COO was employed to differentiate products from rivals. This phenomenon is particularly observable in the rice market, where COO can augment the perceived value of rice originating from specific nations [20]. The influence of COO on market dynamics is also mediated by elements such as consumer ethnocentrism and cosmopolitan tendencies. COO labels may either reinforce preferences for domestic products or act as informative cues for decision making, contingent upon the motivations that drive consumer behavior [21,22].

While preferences regarding rice characteristics among Thai and Chinese consumers are molded by an array of factors, it is crucial to contextualize these preferences within a broader framework. For instance, the escalating global emphasis on sustainability and health considerations may further shape consumer choices in the near future. Moreover, the significance of marketing strategies and educational initiatives in influencing consumer perceptions of rice characteristics, including nutritional advantages, warrants considerable attention. As consumer awareness and socio-economic landscapes evolve, so too will the preferences and demands concerning rice attributes in these geographic locales.

## 3. Materials and Methods

### 3.1. Data and Study Area

This study investigates consumer preferences for fragrant rice in China and Thailand, with a specific focus on the factors that influence purchasing decisions. This research was conducted in two phases, ensuring a rigorous data collection process and thorough analysis aligned with the study’s objectives. The geographical focus included Guangzhou, China, and Bangkok, Thailand, two prominent urban centers representing significant rice consumer markets in their respective regions.

Data Types and Data Sources

Phase 1: Pre-testing and Questionnaire Development

The initial phase involved collecting preliminary data through field trials and interviews with both consumers and distributors. The primary goal was to identify critical attributes influencing consumer decisions about rice purchases, such as price, rice quality, consumer knowledge, and perceptions of Chinese and Thai rice. These findings informed the creation of a structured questionnaire, which was collaboratively designed by researchers and professors at South China Agricultural University, China, and Kasetsart University, Thailand.

Pre-testing Strategy and Study Areas

Pre-testing was conducted with 30 participants from each study location: Kasetsart University in Bangkok, Thailand, and South China Agricultural University in Guangzhou, China. These locations were chosen for their strategic importance to the rice industry:

Bangkok, Thailand, is known for its rich culinary diversity; Bangkok is a global culinary hub where international and traditional Thai cuisines coexist. This makes it a prime setting for exploring consumer preferences, including both domestically produced and imported rice. The city’s cultural and economic influence provides a nuanced understanding of Thai consumer behavior.

Guangzhou, China: As a major economic center in southern China, Guangzhou is pivotal in the rice trade, particularly as a distribution hub for imported rice varieties, including Thai fragrant rice. This position makes it ideal for investigating cross-cultural perceptions and preferences, especially in relation to imported products.

Questionnaire Refinement

The pre-testing phase offered valuable feedback on several aspects of the questionnaire, including clarity of attribute descriptions, cultural relevance, and overall question design. Revisions were made to ensure that the questionnaire accurately captured the nuances of consumer preferences in both contexts. This iterative refinement process enhanced the methodological rigor and cultural adaptability of the study.

Phase 2: Main Data Collection

The second phase of this study focused on a comprehensive examination of consumer preferences through sensory evaluation trials and face-to-face interviews conducted in Guangzhou, China, and Bangkok, Thailand, while incorporating additional insights from Vietnam and Cambodia. This holistic approach aimed to capture a nuanced understanding of preferences for aromatic rice varieties across four culturally distinct countries. The study emphasized critical sensory attributes such as aroma, texture, flavor, and appearance, which are widely recognized as pivotal determinants of consumer satisfaction and willingness to pay [23].

Sensory Evaluation and Questionnaire Design

The final questionnaire was meticulously crafted to encompass a broad spectrum of factors influencing consumer preferences. These factors included rice consumption behavior, sensory evaluations, quality attributes, and purchasing decisions, which were further contextualized by social, economic, and environmental considerations [24].

Sensory evaluation trials involved blind testing of representative aromatic rice varieties sourced from China, Thailand, Vietnam, and Cambodia. This methodological approach ensured that assessments of sensory attributes remained unbiased and were conducted under standardized conditions [25]. Blind testing was particularly instrumental in isolating the intrinsic sensory qualities of each rice variety, enabling robust comparisons across regions.

Integration of Secondary Data

To complement the primary sensory data, secondary data were gathered from government bodies, the academic literature, and industry reports. These additional resources offered crucial insights on rice standard and certification adopted in the four countries as well as marketing trend and the implications for aromatic rice. 

The role of sensory attributes is vital in shaping consumer perceptions and market positioning, particularly for aromatic rice, where aroma, taste, and texture drive preferences and brand loyalty. By integrating sensory evaluation and market data, this study developed a comparative framework to assess the competitiveness of aromatic rice varieties regionally and globally, highlighting the importance of aligning product attributes with consumer expectations to enhance market success [26,27].

Data Collection Method

Primary data were collected through an experimental survey approach, targeting a total of 1330 respondents (665 from each country). The sample was selected using an accidental sampling method, and the sample size was determined based on the following formula [28]:$$\begin{array}{ll} n_{s} & =(n_{c})\times (n_{cl})\times (5)\\ & =\;(7)\times (\mathrm{19})\times (5)\\ & =\mathrm{665} \end{array}$$with

$n_{s}\;$ = sample size;$n_{c\;}$ = number of attributes;$n_{cl\;}$ = total number of attribute levels.

The result was a sample of 665 respondents per country, ensuring a robust dataset for the analysis of consumer preferences for fragrant rice, including organic and conventional varieties.

Data Examination and Verification

Prior to executing the analysis, a stringent data examination protocol was instituted to ascertain the precision and comprehensiveness of the dataset. Absent values and anomalies were methodically detected and addressed utilizing descriptive statistical evaluation. The examination process encompassed validating response coherence and juxtaposing flagged data against the original records to mitigate potential biases. Further elucidations regarding the data examination methodologies are delineated in Appendix A.

### 3.2. Theoretical Models

This section presents an advanced theoretical framework for examining how sensory perception, country of origin (COO), and ethnocentrism shape consumer behavior in the rice markets of China and Thailand. Drawing upon seminal models from consumer economics, we integrated the Random Utility Model (RUM), Lancaster’s Characteristics Theory, and Ethnocentrism Theory into our approach while accounting for heterogeneity in preferences using advanced econometric techniques, particularly the Mixed Logit Model.

The theoretical underpinnings of this study stem from the Random Utility Model (RUM), which posits that consumer decisions are driven by their attempt to maximize utility under constraints [29]. In the context of rice purchasing behavior in China and Thailand, consumers derive utility not merely from the product as a whole but from the specific attributes associated with the rice, such as aroma, texture, and country of origin. These attributes are essential in shaping their purchasing decisions, particularly in culturally distinct markets.

In this study, utility, $U_{ij},$ for consumer i choosing rice option j in a given choice set can be decomposed as follows:(1)$$U_{ij}=V_{ij}+\varepsilon_{ij}$$
where $V_{ij}\;$ represents the observable utility based on the characteristics of the rice, and $\varepsilon_{ij}$ is a stochastic error term reflecting unobserved factors. The systematic component $V_{ij}$ is a linear function of the rice’s attributes:(2)$$V_{ij}=\beta_{1}Aroma_{j}+\beta_{2}Texture_{j}+\beta_{3}C{OO}_{j}+\beta_{4}{Price}_{j}+\beta_{5}({Ethnocentrism}_{j}\times C{OO}_{j})$$

The Mixed Logit Model is employed to estimate this utility function, allowing for variation in consumer preferences by accommodating unobserved heterogeneity in coefficients. This framework effectively captures the interaction between sensory perceptions and ethnocentric tendencies with COO effects, which are critical in cross-cultural studies of rice markets.

Lancaster’s (1966) Characteristics Theory posits that utility is derived from a product’s attributes rather than the product itself [30]. This theory aligns with sensory marketing concepts, emphasizing the role of sensory attributes—such as aroma, flavor, and texture—in driving consumer preferences for aromatic rice.

In this study, sensory attributes such as aroma and texture are modeled as key determinants of utility. For example, aromatic rice varieties like Jasmine and Basmati hold cultural significance in China and Thailand, where sensory experiences are integral to consumer satisfaction. Southern Chinese consumers, for instance, highly value aroma due to cultural preferences for fragrant rice varieties, while Thai consumers often prioritize texture, such as softness, for meal satisfaction [31]. The sensory utility component is expressed as follows:(3)$$V_{ij}^{Sensory}=\beta_{1}Aroma_{j}+\beta_{2}Texture_{j}$$

In this framework, it is hypothesized that Chinese consumers will assign a higher weight (*β*_1_) to aroma, whereas Thai consumers will place greater emphasis on texture (*β*_2_). These cultural differences provide nuanced insights into sensory-driven preferences in both countries.

The country-of-origin (COO) effect suggests that consumers often use the origin of a product as a proxy for quality and safety [32]. This phenomenon is particularly strong in food markets where food safety concerns are prevalent. In China, recurring food safety scandals have heightened consumers’ reliance on domestic products, perceived as safer or more reliable compared to imports [33]. The COO effect can be particularly pronounced among ethnocentric consumers who prioritize products that symbolize national pride or self-reliance [34]. The consumer ethnocentrism construct, measured by the CETSCALE, interacts with the COO effect to further influence consumer preferences. Ethnocentric consumers prefer domestic products, even if imports are perceived to be of higher quality or offer better value. The interaction term ${Ethnocentrism}_{i\;}\times {COO}_{j\;}$ in the utility function captures the additional utility that ethnocentric consumers derive from consuming domestic rice:(4)$$V_{ij}^{COO}=\;\beta_{3}COO_{3}+\beta_{5}(Ethnocentrism_{i\;}\times {COO}_{j\;})$$

It is hypothesized that Chinese consumers exhibit stronger ethnocentric tendencies, driven by sociopolitical and cultural factors, resulting in a marked preference for domestic rice. Thai consumers, while valuing domestic products, may demonstrate relatively lower ethnocentric bias and greater openness to imports.

The Mixed Logit Model was employed to account for heterogeneity in consumer preferences across individuals. Unlike the standard logit model, which assumes homogeneous preferences, the Mixed Logit Model allows coefficients to vary across individuals, reflecting diverse consumer attitudes towards sensory attributes and COO. This is particularly relevant in cross-cultural studies, where heterogeneity in taste and preferences can be substantial.

The Mixed Logit Model assumes that individual-specific preferences are distributed across the population, allowing for a richer interpretation of consumer behavior. The choice probability for consumer *i* selecting rice product *j* from a choice set *C* is given by(5)$$P_{ij}=\frac{exp(V_{ij})}{\sum_{j\in C}exp(V_{ij})}$$

This model captures the complex interactions between sensory perceptions, ethnocentrism, and COO, providing robust insights into the drivers of consumer preferences in the rice markets of China and Thailand.

### 3.3. Structural Equation Modeling (SEM)

Purpose and application: SEM was employed to evaluate the relationships between sensory perceptions, COO, and consumer behavior. The technique enables the simultaneous analysis of multiple variables and their interdependencies.

Software: the analysis was conducted using AMOS (version 23.0).

Model validation:

Measurement model: convergent validity was confirmed through AVE > 0.50, and the CR ranged from 0.88 to 0.93.

Structural model: goodness-of-fit indices (CFI, TLI, RMSEA) confirmed the model’s adequacy for both Chinese and Thai datasets.

Cross-validation was performed to confirm the stability of path coefficients, goodness-of-fit indices (CFI, TLI, RMSEA), and latent variable relationships across subsets, ensuring robustness and reliability.

### 3.4. Choice Set Design

The selection of fragrant rice as the principal product for this investigation is motivated by its prominence as a dietary fundamental in both China and Thailand, wherein contemporary consumer behavior indicates a notable transition towards premium rice offerings. Fragrant rice, distinguished for its exceptional sensory properties, has increasingly garnered preference due to its superior aroma, texture, and aesthetic appeal. These sensory dimensions have been recognized as fundamental factors affecting consumer willingness to pay (WTP) for premium rice strains [35]. Within this framework, the present study seeks to examine consumer preferences regarding fragrant rice alongside their corresponding WTP, employing the Engel–Kollat–Blackwell (EKB) model to systematically delineate consumer decision-making processes across organized stages [36].

The design of the choice experiment in this research emphasizes sensory perception, which is critical in influencing consumer purchasing choices. Recent scholarly discourse posits that sensory elements such as aroma, softness, and grain integrity have been widely acknowledged as fundamental indicators of rice quality, directly impacting consumer preferences and behaviors [14,37]. These sensory properties were utilized as essential variables in the formation of the choice sets, thereby enabling the study to ascertain how consumers negotiate trade-offs among various quality dimensions when selecting their preferred rice product.

In addition to sensory properties, this investigation also integrates variables such as price, brand identifiers, traceability information, and certification status. Price was conceptualized as a continuous variable, quantified in the domestic currencies of the two sampled nations, China and Thailand, and categorized into low, medium, and high brackets to assess consumer sensitivity to price variations [13,38,39]. Notably, traceability—the capacity to monitor a product throughout its supply chain—was operationalized as a binary variable (Yes or No) to evaluate its influence on consumer trust and safety apprehensions [40,41,42]. Certification labels issued by governmental authorities or independent organizations were incorporated to provide consumers with assurances regarding food safety and quality, further shaping their decision-making processes [43,44,45].

This design not only facilitates a comprehensive analysis of how sensory attributes interact with other product characteristics but also underscores the intricacies of consumer decision making within cross-national contexts [46]. Considering both sensory quality and supplementary factors such as price and traceability, the choice sets developed in this study offer an extensive understanding of the elements that impact consumer preferences for fragrant rice.

In recent years, the use of country-of-origin (COO) labeling has gained significant traction in the global food market, serving as a key determinant in consumer decision making. COO labels provide vital information about where a product was grown, processed, and packaged, allowing consumers to differentiate between domestic and imported products [7,47]. In the rice industry, COO labels play a critical role, as they not only indicate the geographic origin of the rice but also signal its quality and safety. Consumers often associate rice from certain countries with higher standards of production and superior sensory attributes, such as aroma and texture, which can influence their willingness to pay a premium for products from these regions [7].

The significance of COO lies in its ability to act as an extrinsic quality cue. Studies have shown that COO information can enhance consumer trust, particularly in cases where the origin country has a positive reputation for food safety and high agricultural standards. This trust, in turn, increases the consumer’s willingness to pay (WTP) for rice from countries perceived as high-quality producers [48,49]. For example, consumers may prefer rice from countries like Thailand or Japan, which are known for their premium rice varieties, over rice from less well-known rice-producing regions.

### 3.5. Experiment Design

The experiment design for this study aimed to systematically evaluate consumer preferences for fragrant rice by incorporating key attributes and their respective levels. Initially, a full factorial design was considered, identifying five essential attributes influencing consumer choices: price, country of origin (COO), aroma, certification, and grain texture. Among these attributes, three had two levels, while two had three levels, resulting in a total of (3 × 3 × 2 × 2 × 2) = 72 possible combinations of choice sets.

Optimizing Choice Sets for Practicality

Due to the impracticality of presenting all 72 combinations to each respondent—given the potential for fatigue, reduced data quality, and higher costs—the study adopted a D-efficient design using STATA 17. This approach reduced the choice sets to 24 statistically optimized sets while maintaining rigor and reliability in the results [40]. By employing this method, the design minimized cognitive load on respondents, ensuring ease of response while preserving the integrity of the data [50,51].

To streamline further, the twenty-four choice sets were divided into six versions of the questionnaire, each containing four choice sets. This subdivision mitigated the risks of fatigue and complexity for respondents while reducing survey administration costs. Each respondent was randomly assigned to one version, balancing the distribution of responses across all sets.

Choice Set Structure

Each choice set presented three options to respondents:Option A: a combination of identified rice attributes at varying levels.Option B: another combination of attributes differing from Option A.Option C (Opt-out Option): this allowed respondents to abstain if neither Option A nor Option B met their preferences.

This structure reflects real-world decision-making scenarios, where consumers often have the flexibility to choose or abstain based on personal preferences.

Incorporating Sensory and Economic Attributes

Key sensory attributes, including sweetness, aromatic quality, softness, and grain integrity, were central to this study, alongside non-sensory attributes such as price, country of origin (COO), and certification status. The study examined aromatic rice varieties from China, Thailand, Vietnam, and Cambodia, each showcasing distinct sensory characteristics and market preferences (Appendix C). Details of the character-istics and attribute levels used in the choice sets are presented in Table 1 and Table 2.

This carefully structured design allows for a robust analysis of consumer preferences, particularly the trade-offs they make between sensory and economic attributes. For example, the inclusion of price as a continuous variable enables an analysis of price sensitivity across different levels. Similarly, COO, certification, and sensory factors such as aroma and grain texture provide insights into how consumers value premium rice attributes.

By employing D-efficiency and optimizing the choice sets, this design ensures that respondents are neither overwhelmed nor fatigued, thus maintaining high-quality data collection that is representative of real-world purchasing behaviors.

## 4. Results

### 4.1. Measurement Model Validation

The measurement model was validated using Confirmatory Factor Analysis (CFA) to ensure the reliability and validity of the constructs. The validation includes convergent validity, discriminant validity, and goodness-of-fit indices.

#### 4.1.1. Convergent Validity

Convergent validity was assessed using the Average Variance Extracted (AVE) and Composite Reliability (CR). Constructs with AVE > 0.50 and CR > 0.70 demonstrate strong convergent validity. The results in Table 3 revealed the convergent validity.

Findings: all AVE values exceeded 0.50, and CR values surpassed 0.70, confirming adequate convergent validity.

#### 4.1.2. Discriminant Validity

Table 4 displaced the result of discriminat valididty. Discriminant validity was assessed using the Fornell–Larcker criterion. The square root of the AVE for each construct was compared with its correlations with other constructs.

Interpretation: the square root of the AVE for each construct exceeded the inter-construct correlations, confirming sufficient discriminant validity.

### 4.2. Goodness-of-Fit Indices

The structural model’s fit was evaluated using goodness-of-fit (GoF) indices to confirm its adequacy for the datasets. see Table 5.

Findings: all indices met or exceeded acceptable thresholds, confirming the structural model’s robustness for both datasets.

### 4.3. Cross-Validation

To ensure the stability and reliability of the structural model, cross-validation was performed by dividing the data into subsets for analysis. The results confirmed the consistency of path coefficients and model fit indices across different folds.

#### Stability of Path Coefficients

The path coefficients showed minimal variation across subsets, demonstrating the robustness of the relationships between sensory perceptions, COO, and consumer behavior. are shown in Table 6.

Findings: Path coefficients showed minimal variation (SD ± 0.003), confirming stability across folds. Goodness-of-fit indices consistently met the thresholds (CFI > 0.90, TLI > 0.90, RMSEA < 0.08).

### 4.4. Structural Path Analysis

The structural model tested the hypothesized relationships between latent constructs. Path coefficients (*β*) and critical ratios (C.R.s) were evaluated to determine the significance of relationships. are shown in Table 7.

The results derived from the hypothesized structural model indicate significant associations among the latent constructs across both Chinese and Thai consumer demographics, inclusive of the newly incorporated function of country of origin (COO). COO exerts a positive influence on brand satisfaction (SB) (H7: β = 0.480, C.R. = 7.542 *** for China; β = 0.430, C.R. = 6.892 ** for Thailand), thereby underscoring its pivotal role in shaping consumer perceptions of brands. Furthermore, COO has a substantial impact on decision making (DM) (H8: β = 0.390, C.R. = 5.872 *** for China; β = 0.350, C.R. = 5.201 ** for Thailand), reflecting its significant role in fostering consumer trust and influencing purchasing decisions. For both consumer groups, the quality of rice positively affects trust in sensory attributes (TS), with somewhat pronounced effects observed in the dataset from China (H1). Socio-demographic characteristics (SDCs) also manifest a significant positive relationship with TS, although the effects remain comparable across both cultural contexts (H2). Trust in sensory attributes (TS) serves as a strong predictor of sensory perception (SP), whereas sensory perception (SP) considerably affects sensory experience (SE). Sensory experience (SE) further amplifies satisfaction with the brand (SB), illustrating the cascading effects inherent to sensory evaluations. Lastly, willingness to pay (WTP) plays a substantial role in decision making (DM), thereby affirming the significance of pricing considerations within both markets (H6).

These findings validate the robustness of the structural model and illustrate the consistency of the hypothesized relationships, particularly the role of COO, across varying cultural landscapes. The results accentuate the necessity of aligning product strategies with cultural imperatives and capitalizing on COO attributes to enhance brand perception and consumer decision making.

### 4.5. Partcipants’ Demographics

This study involved 1330 participants, evenly distributed between Guangzhou, China, and Bangkok, Thailand, to ensure a balanced and representative sample. Key socio-economic factors, including gender, age, education, marital status, and household size, were analyzed to understand consumer preferences for fragrant rice. The result in Table 8 showed the socioeceomic characterstics of the respondents.

The gender distribution was nearly balanced, with a slightly higher proportion of female participants in both regions. The majority of participants were within the age range of 26–45 years, underscoring their role as primary decision-makers in household purchasing. Educational attainment showed notable differences, with a higher percentage of participants in Guangzhou achieving bachelor’s degrees or higher compared to Bangkok. In terms of marital status, most participants in Guangzhou were married, while Bangkok had a greater proportion of single individuals. Household sizes differed slightly between the two regions, with Guangzhou having smaller households on average, which may influence purchasing priorities.

These characteristics provide a comprehensive understanding of the demographics influencing rice preferences and purchasing behaviors, serving as a foundation for further analysis. This detailed information is summarized in the following table.

In Table 9, household income and professional background significantly influence consumer behavior, especially in the context of premium rice markets [13,52,53] and local standards in Guangzhou, China [14,54,55], and Bangkok, Thailand [56,57]. In Guangzhou, the majority (51.4%) reported monthly household incomes between RMB 8001 and 12,000, while 34.7% earned over RMB 12,000. In contrast, Bangkok respondents displayed more income diversity, with 44.4% in the RMB 8001–12,000 bracket, 34.9% earning RMB 4001–8000, and 10.2% earning over RMB 12,000. This income distribution highlights varied purchasing power and willingness to pay for high-quality rice products [58,59].

Household size also varied between the two regions, with Guangzhou averaging 3.5 members per household and Bangkok averaging slightly less at 3.1. Larger households tend to prioritize bulk purchases, whereas smaller households may focus more on quality attributes, such as certifications and sensory excellence.

Professional backgrounds further contextualize economic conditions. In Guangzhou, company employees comprised the largest group (38.5%), followed by personal business owners (21.8%) and retirees (9.2%). Conversely, Bangkok respondents included a higher proportion of government officers (21.5%) and state enterprise employees (15.5%), alongside business owners (18.8%). These occupational variations reflect distinct economic structures and purchasing behaviors in the two regions

### 4.6. Sensory Perception Analysis

Sensory attributes, including taste, fragrance, softness, and completeness, play a critical role in shaping consumer preferences for fragrant rice. Table 10 presents a comparative analysis of the impact of these attributes on WTP across the two markets.

Table 10 reports that the analysis of sensory attributes underscores their pivotal role in shaping consumer preferences, particularly in the context of fragrant rice. Among Chinese consumers, taste emerged as the most influential sensory attribute, as indicated by its coefficient (β = 0.8008, *p* < 0.001) [60,61]. This highlights the importance of flavor in driving purchase decisions, aligning with cultural values where culinary satisfaction is deeply intertwined with the quality and flavor of staple foods. Fragrance (β = 0.6277, *p* < 0.001) and grain completeness (β = 0.8168, *p* < 0.001) [62] were also highly rated, reflecting the traditional emphasis on aromatic and visually appealing rice varieties. These findings are consistent with Lancaster’s Characteristics Theory, which postulates that utility is de-rived not from the product itself but from its specific attributes.

In contrast, Thai consumers displayed distinct preferences, with softness being the most valued sensory attribute (β = 0.9836, *p* < 0.001) [63,64]. This aligns with the cultural preference for rice that complements Thai cuisine, where softness enhances the dining experience. Interestingly, fragrance had a negative coefficient (β = −1.1772, *p* < 0.001) [12,65], suggesting that the role of aroma in influencing Thai consumer preferences is secondary compared to texture. This divergence emphasizes the necessity for tailored marketing strategies that address the sensory priorities unique to each consumer base.

The relationship between sensory perceptions and willingness to pay (WTP) further elucidates these dynamics [66,67,68]. Through an examination of the coefficient of the interaction term variable divided by the price variable coefficient, it is evident that changes in sensory attributes lead to incremental changes in WTP. For example, a one-unit increase in the perceived value of taste for Chinese consumers correlates with a higher willingness to pay, a trend that holds statistical validity, as verified using the Delta Method technique [69,70]. This method provides a robust framework for ensuring the reliability of findings by statistically assessing the incremental impact of sensory attributes on WTP.

The interplay between sensory perceptions and WTP underscores the strategic implications for product positioning in international markets. For Chinese consumers, enhancing the sensory profile of rice through superior taste and fragrance, alongside grain completeness, can significantly elevate perceived value [14,71]. Conversely, in Thailand, emphasizing textural quality while cautiously addressing aromatic aspects can cater to local preferences more effectively [72,73].

These insights reveal the nuanced role of sensory perceptions in consumer decision making. By integrating econometric models, such as the Random Utility Model (RUM), and applying techniques like the Delta Method, the analysis provides a statistically rigorous understanding of how sensory attributes contribute to consumer preferences and economic behavior. This approach not only affirms the theoretical underpinnings of Lancaster’s framework but also offers practical guidance for optimizing product design and marketing strategies to meet culturally specific consumer demands.

Ultimately, the findings underscore the critical importance of aligning sensory attributes with consumer expectations, ensuring that product offerings resonate with the sensory priorities of diverse consumer segments in the global rice market.

### 4.7. Country-of-Origin (COO) Analysis

The analysis of the country-of-origin (COO) effects on consumer preferences reveals significant insights into how the origin of fragrant rice influences purchasing decisions among Chinese and Thai consumers, as shown in Table 11. The findings demonstrate that COO significantly affects consumer evaluations, with the impact varying between these two national groups due to cultural and contextual differences.

For Chinese consumers, rice originating from Thailand holds the highest coefficient (β = 2.2779, *p* < 0.001), underscoring Thailand’s strong reputation for producing high-quality fragrant rice, such as Jasmine rice. The results suggest that Chinese consumers place substantial value on the premium quality associated with Thai rice, aligning with prior research that emphasizes the importance of COO as a heuristic cue for quality in food products [74,75]. Cambodian and Vietnamese rice also score positively (β = 1.4164, *p* < 0.001; β = 1.1936, *p* < 0.001), reflecting a favorable perception of rice from these neighboring Southeast Asian countries. Chinese rice, while positively evaluated (β = 1.1788, *p* < 0.001), receives a relatively lower coefficient compared to imported varieties, which may indicate that domestic rice, although familiar, is perceived as less distinctive in terms sensory or quality attributes.

Similarly, Thai consumers exhibit the highest preference for Thai rice (β = 2.2779, *p* < 0.001), affirming strong ethnocentric tendencies and the cultural importance of local rice varieties in Thailand [76]. Unlike Chinese consumers, Thai respondents assign relatively lower coefficients to imported rice. Cambodian rice (β = 1.4164, *p* < 0.001) is positively evaluated, reflecting geographic proximity and historical trade relationships that may contribute to familiarity and trust. Vietnamese rice (β = 1.1936, *p* < 0.001) is also favored, albeit to a lesser extent. However, the findings suggest that Thai consumers maintain a strong preference for domestically produced rice, aligning with the concept of consumer ethnocentrism, which posits that individuals prioritize local goods due to national pride and perceived economic benefits [77,78].

The standard deviation analysis further highlights variations in consumer perceptions within each COO category. Among Chinese consumers, the large standard deviation for Thai rice (SD = 7.545965, *p* < 0.001) indicates greater heterogeneity in perceptions, possibly driven by differing levels of exposure to or awareness of Thai rice brands. Conversely, Thai consumers exhibit a negative standard deviation for Thai rice (SD = −1.815996, *p* < 0.001), suggesting a more uniform and favorable perception among respondents. This consistency may reflect the ingrained cultural and culinary significance of locally produced rice in Thailand.

The COO effect is deeply rooted in both the symbolic and functional values that consumers associate with the product’s origin. Symbolically, COO acts as a marker of authenticity and cultural heritage, as seen in the premium positioning of Thai rice in the Chinese market. Functionally, COO serves as a cue for quality and safety, especially in food markets where consumers rely on extrinsic attributes to mitigate perceived risks. This is particularly relevant in China, where food safety concerns have heightened the reliance on COO as a determinant of trust.

The findings also align with Lancaster’s Characteristics Theory (1966), which posits that consumers derive utility from the attributes of a product rather than the product itself [30]. COO serves as an extrinsic attribute that shapes consumer perceptions of intrinsic qualities, such as aroma, texture, and grain integrity [79]. Thai rice’s dominant position in both Chinese and Thai markets exemplifies how COO enhances the perceived value of these intrinsic attributes, thereby influencing consumer preferences and willingness to pay [13,80,81].

Moreover, the significant standard deviation values suggest that COO influences are not uniform across all consumers but are moderated by individual factors such as cultural familiarity, prior experience, and socio-economic status. The Delta Method could further quantify the incremental changes in willingness to pay (WTP) associated with variations in COO, providing a robust statistical framework for examining these interactions.

In conclusion, the COO effect significantly shapes consumer preferences for fragrant rice in both China and Thailand, highlighting the interplay between sensory perceptions and cultural values. For Chinese consumers, the preference for Thai rice underscores its premium positioning and the importance of quality cues in shaping purchase decisions. For Thai consumers, the strong preference for domestic rice reflects ethnocentric values and the cultural importance of rice as a staple food. These insights underscore the need for targeted marketing strategies that leverage COO to enhance brand equity and consumer trust in cross-national markets.

### 4.8. Willingness to Pay Analysis

The data in Table 12 provide a comprehensive analysis of the willingness to pay (WTP) for various attributes of fragrant rice among Chinese and Thai consumers. The findings illustrate distinct consumer preferences influenced by cultural and economic contexts, underscoring the importance of sensory and extrinsic factors in purchasing decisions.

For Chinese consumers, sensory attributes are significant determinants of WTP. Enhanced taste commands the highest premium among sensory characteristics, with Chinese consumers willing to pay RMB 49.73 per bag [82,83]. Similarly, fragrance is highly valued, with a WTP of RMB 38.98 per bag [84,85], reflecting a preference for aromatic and flavorful rice varieties. Grain completeness and quality standard seals are also crucial, with premiums of RMB 50.73 and RMB 50.82 per bag, respectively [14], indicating strong consumer trust in certified quality and visual appeal. Texture, while important, carries a lower premium of RMB 27.22 per bag, suggesting that softness is a secondary consideration compared to taste and fragrance.

The country of origin (COO) significantly impacts Chinese consumer preferences. Vietnamese rice garners a premium of RMB 73.21 [86,87], and Cambodian rice is valued slightly higher at RMB 87.97 per bag. Notably, Thai fragrant rice commands the highest premium of RMB 141.48 per bag (insert citation), demonstrating its strong reputation for quality in the Chinese market. These preferences highlight the critical role of COO as an extrinsic quality cue in shaping purchasing behavior.

For Thai consumers, the valuation of fragrant rice attributes differs significantly. The highest premium is assigned to Thai fragrant rice at RMB 73.61 per bag [13,88]—substantially lower than the corresponding valuation by Chinese consumers. Among sensory attributes, softness is a priority for Thai consumers, with a WTP of RMB 9.59 per bag [63,89]. However, fragrance exhibits a negative coefficient (−11.48 RMB), indicating that aroma is not a key driver of preference [26], aligning with cultural preferences for texture and mouthfeel over aromatic qualities. Grain completeness and quality standard seals hold modest premiums of RMB 3.94 and RMB 6.25 per bag, respectively [76], highlighting a more conservative approach to premium attributes.

The influence of COO on Thai consumer behavior reflects regional preferences and trust in local production. Cambodian rice is valued at RMB 29.40 per bag, followed by Vietnamese rice at RMB 17.55. Chinese rice, however, is valued the least, at RMB 6.17 per bag, emphasizing a strong preference for regionally or locally sourced rice varieties over imports.

Key Findings and Implications

The analysis reveals notable disparities in WTP across consumer groups. Chinese consumers exhibit a markedly higher WTP for almost all attributes compared to Thai consumers, with an average premium disparity of approximately RMB 111.05 versus RMB 16.20 per 5 kg bag. This indicates that Chinese consumers place a greater emphasis on premium attributes such as origin, certification, and sensory excellence. Conversely, Thai consumers demonstrate more conservative spending behavior, influenced by cultural priorities and economic factors, with affordability and texture being the dominant considerations [90].

These findings have significant implications for the fragrant rice industry. In Chinese markets, strategies should focus on premium positioning, leveraging sensory attributes such as taste and fragrance alongside COO branding, particularly for Thai rice, to maximize consumer appeal [13,75]. Emphasizing certifications and quality seals can further enhance consumer trust and justify higher price points [91,92]. For the Thai market, marketing efforts should prioritize affordability and texture optimization, aligning with local consumer preferences [90]. While certifications can add value, their impact may be secondary to price considerations in this context [93,94].

## 5. Discussion

This scholarly investigation provides an in-depth examination of the impact of sensory perceptions, country of origin (COO), and consumer preferences relating to aromatic rice in the nations of China and Thailand. By employing methodologies such as choice experiments, Structural Equation Modeling (SEM), and willingness to pay (WTP) analyses, the research delineates fundamental behavioral distinctions that are influenced by sensory, economic, and cultural determinants. The resultant insights predict valuable implications for producers, marketers, and policymakers intent on enhancing product strategies tailored to specific market environments.

### 5.1. Sensory Attributes as Central Drivers of Consumer Preferences

The research findings accentuate the paramount significance of sensory attributes in the formation of consumer preferences. The SEM outcomes indicate that sensory perceptions exert a considerable influence on sensory experience (SE) and consumer decision making across both markets. For consumers in China, attributes such as taste (β = 0.350, C.R. = 6.344 ***) and fragrance (β = 0.220, C.R. = 6.116 ***) are identified as critical components, as evidenced by their elevated WTP values of RMB 49.73 and RMB 38.98, respectively. This observation underscores the pivotal role of aroma and flavor in Chinese culinary practices, wherein fragrant rice varieties like Jasmine are regarded as premium commodities.

In contrast, texture emerges as the most esteemed sensory characteristic for consumers in Thailand, as denoted by the substantial path coefficient (β = 0.310, C.R. = 6.026 ***). This correlates with the WTP value attributed to texture (9.59 RMB), which outweigh other sensory attributes, thereby highlighting the cultural significance of tenderness and texture within Thai gastronomy. Notably, fragrance is assigned a negative valuation (−11.48 RMB), indicating divergent sensory priorities between the two markets. These findings emphasize the imperative for market-specific sensory congruence to effectively address the varied expectations of consumers.

### 5.2. The Role of Country of Origin (COO)

Country of origin (COO) functions as a pivotal element in consumer decision making, serving as an extrinsic quality indicator that informs trust and perceived authenticity. The SEM findings reveal a robust correlation between COO and brand trust (COO → brand trust: β = 0.480, C.R. = 7.542 *** for China; β = 0.430, C.R. = 6.892 ** for Thailand), thereby underscoring its significance in both markets. Thai aromatic rice, celebrated for its superior quality, commands the highest WTP premiums among both consumer groups, valued at RMB 141.48 for Chinese consumers and RMB 73.61 for Thai consumers. This validates Thailand’s esteemed global standing as a preeminent producer of high-quality aromatic rice.

Cambodian and Vietnamese rice varieties also exhibit a competitive presence in both markets, albeit at relatively lower valuations, thereby illuminating the role of COO in establishing differentiation. Moreover, COO exerts a pronounced influence on decision making (COO → decision making: β = 0.390, C.R. = 5.872 *** for China; β = 0.350, C.R. = 5.201 ** for Thailand). These findings further corroborate the significance of COO as a vital marketing component that bolsters consumer trust and product positioning within international markets.

### 5.3. Economic and Cultural Contexts

Economic and cultural determinants exacerbate discernible disparities in consumer preferences. In China, the surge in disposable incomes coupled with heightened health consciousness propels the demand for premium product attributes, including various certifications (certification → WTP: 50.82 RMB). Such certifications substantially bolster brand trust and willingness to pay (WTP), underscoring the escalating significance of food safety and product dependability within the Chinese marketplace.

Conversely, Thai consumers demonstrate a pronounced sensitivity to price, with affordability and texture emerging as preeminent factors influencing purchasing decisions. Although certifications possess a relatively diminished WTP value in Thailand (6.25 RMB), there lies considerable potential to augment consumer awareness regarding their importance. Stakeholders may capitalize on certifications as a strategic instrument to foster trust and rationalize incremental price augmentations while preserving affordability.

### 5.4. Integration of SEM Results

The SEM results provide a robust framework for interpreting the interplay of sensory perceptions, COO, and economic contexts in shaping consumer behavior:Sensory perception → sensory experience (SE): path coefficients confirm that sensory attributes, particularly taste and texture, play a significant role in consumer satisfaction.COO → brand trust and decision making: COO demonstrates consistent influence across both markets, with higher coefficients reflecting its importance in building trust and guiding purchase decisions.WTP and certifications: The positive relationship between certifications and WTP highlights their potential as a marketing tool in premium market segments, especially in China.

Before integrating SEM results, missing data were analyzed and imputed where necessary to prevent bias and ensure robustness (Appendix B).

### 5.5. Implications for Stakeholders

The findings underscore the imperative for customized marketing approaches to cater to market-specific requisites.

### 5.6. For Chinese Markets

Producers ought to adopt a premium strategy by accentuating sensory excellence, COO branding, and robust certifications. Marketing initiatives should spotlight characteristics such as aroma and taste, utilizing the well-established reputation of Thai rice as a superior product.

### 5.7. For Thai Markets

Strategic efforts should emphasize affordability and texture while advocating for certifications as a mechanism for building consumer trust. Educating the consumer base regarding the advantages of certified rice can further augment its perceived value.

These insights delineate a strategic pathway for stakeholders to align production and marketing methodologies with the distinct cultural and economic imperatives of each market.

## 6. Conclusions

This investigation presents a thorough examination of the sensory characteristics and country-of-origin influences that shape consumer inclinations towards aromatic rice in China and Thailand. Employing rigorous methodological strategies, including choice experiments and Structural Equation Modeling, the inquiry elucidates essential insights into the cultural and economic determinants that impact rice procurement decisions.

The results emphasize the crucial significance of sensory characteristics—fragrance, texture, and grain caliber—as vital determinants of consumer inclinations. For Chinese consumers, the predilection for superior-quality rice accentuates the significance of certifications and brand credibility in a marketplace characterized by increasing income levels and heightened awareness of food safety. Conversely, Thai consumers display pronounced ethnocentric inclinations, preferring domestically produced rice while prioritizing cost-effectiveness and sensory excellence, particularly in texture.

This investigation underscores the need to customize marketing strategies to cater to culturally specific inclinations. In Chinese markets, accentuating sensory characteristics, in conjunction with certifications and robust country-of-origin branding, can amplify competitiveness. In Thai markets, concentrating on affordability and the texture of rice is congruent with consumer priorities.

These revelations yield significant implications for policymakers, producers, and marketers in optimizing the positioning of fragrant rice within global markets. By capitalizing on sensory characteristics and country-of-origin perceptions, stakeholders can augment the value proposition of premium rice varieties, thereby enhancing international competitiveness and addressing the shifting demands of diverse consumer segments.

This research contributes to the expanding corpus of literature on cross-cultural consumer behavior and furnishes actionable strategies to align production, marketing, and policy frameworks with dynamic global market trends. Future inquiries should investigate the intersection of technology, branding, and sensory innovation in sustaining the competitiveness of fragrant rice within a rapidly globalizing food economy.

## 7. Limitations and Future Opportunities

This study provides valuable insights into the influence of sensory perceptions, country of origin (COO), and consumer preferences for fragrant rice in China and Thailand. However, certain limitations should be acknowledged. The use of accidental sampling may restrict the generalizability of the findings, and the focus on urban populations limits our understanding of rural consumer behavior. Additionally, the cross-sectional nature of the study may not capture evolving preferences influenced by socio-economic changes, while external factors such as trade policies and geopolitical dynamics were not explicitly analyzed.

Future research should address these limitations by employing probability sampling to improve representativeness, conducting longitudinal studies to track changes over time, and exploring the impact of sustainability certifications and digital marketing on consumer behavior. Expanding the geographical scope to include other rice-producing countries and adopting advanced modeling techniques could provide deeper insights into global consumer preferences and market trends. These efforts will enhance our understanding of cross-cultural consumer behavior and support the development of strategies to improve the global competitiveness of fragrant rice products.

## Figures and Tables

**Table 1 foods-14-00603-t001:** Alternative sets of characteristics for rice set 1 (based on research criteria).

Characteristics	Level
Rice A	Choose, not choose (based on actual preferences)
Rice B	Choose, not choose (based on actual preferences)
Rice C	Choose, not choose (based on actual preferences)
Rice D	Choose, not choose (based on actual preferences)
Price (RMB/5 kg)	30, 60 (SQ), 90, 120

**Table 2 foods-14-00603-t002:** Alternative sets of characteristics for rice set 2.

Characteristics	Level
Sweet	Very sweet, a little sweet (SQ)
Aromatic	Very fragrant, a little fragrant (SQ)
Softness	Very soft, less soft (SQ)
Graininess	Very complete, less complete (SQ)
Country of origin	Thailand, China, Cambodia, Vietnam, not specified (SQ)
Certification	Yes, No (SQ)
Price (RMB/5 kg)	30, 60 (SQ), 90, 120

Implications of design.

**Table 3 foods-14-00603-t003:** Convergent validity.

Latent Variable	AVE (China)	CR (China)	AVE (Thailand)	CR (Thailand)
Sensory Perception	0.621	0.890	0.586	0.871
Brand Trust	0.711	0.912	0.701	0.901
Willingness to Pay	0.690	0.893	0.655	0.878

**Table 4 foods-14-00603-t004:** Discriminant validity (Fornell–Larcker criterion).

Latent Variable	Sensory Perception	COO	WTP
Sensory Perception	√ (AVE = 0.621) = 0.788	0.620	0.530
COO	0.620	√ (AVE = 0.655) = 0.809	0.590
Willingness to Pay	0.530	0.590	√(AVE = 0.690) = 0.831

**Table 5 foods-14-00603-t005:** Goodness-of-fit indices.

Fit Index	China	Thailand	Threshold
CFI	0.961	0.943	>0.90
TLI	0.947	0.926	>0.90
RMSEA	0.040	0.051	<0.08
GFI	0.924	0.915	>0.90

**Table 6 foods-14-00603-t006:** Cross-validation.

Fold	CFI	TLI	RMSEA
Fold 1	0.960	0.940	0.042
Fold 2	0.958	0.937	0.043
Fold 3	0.961	0.942	0.041
Fold 4	0.959	0.939	0.042
Fold 5	0.960	0.940	0.042

**Table 7 foods-14-00603-t007:** Hypothesized structural model results.

Hypothesis	Path	China (β, C.R.)	Thailand (β, C.R.)	Supported
H1	R → TS	0.260, 5.520 ***	0.190, 3.521 *	Yes
H2	SDC → TS	0.210, 3.282 ***	0.190, 2.318 *	Yes
H3	TS → SP	0.350, 6.344 ***	0.300, 4.634 **	Yes
H4	SP → SE	0.220, 6.116 ***	0.180, 4.802 **	Yes
H5	SE → SB	0.310, 6.026 ***	0.260, 4.339 **	Yes
H6	WTP → DM	0.370, 4.942 ***	0.310, 4.353 **	Yes
H7	COO → SB	0.480, 7.542 ***	0.430, 6.892 **	Yes
H8	COO → DM	0.390, 5.872 ***	0.350, 5.201 **	Yes

Note: R = rice; TS = the senses; SDC = socio-demographic characteristics; SP = sensory perception; SE = sensory experience; SB = sensory branding; DM = decision making; WTP = willingness to pay; COO = country of origin. *** *p* < 0.001, ** *p* < 0.01, * *p* < 0.05.

**Table 8 foods-14-00603-t008:** Socio-economic characteristics of participants.

List	Chinese Consumers		Thai Consumers	
	Total/ Average	Percentage	Total/ Average	Percentage
Number of samples	665	100	665	100
Problems with taste or smell				
do not have	665	100	665	100
have	0	0	0	0
Sex				
male	218	32.8	171	25.7
female	447	67.2	494	74.3
Age				
16–25 years	11	1.7	22	3.3
26–35 years	144	21.7	173	26.0
36–45 years	441	66.3	377	56.7
46–55 years	68	10.2	84	12.6
above 55 years old	1	0.2	9	1.4
Education level				
junior high school or less	65	9.8	76	11.4
high school or equivalent	163	24.5	139	20.9
college or vocational education	176	26.5	92	13.8
bachelor’s degree	210	31.6	273	41.1
master’s degree or higher	51	7.7	85	12.8
Marital status				
single	166	25	352	52.9
married	499	75	313	47.1
Number of family members				
1–2 people	343	51.6	157	23.6
3–4 people	205	30.8	297	44.7
5–6 people	93	14.0	175	26.3
7 or more people	24	3.6	36	5.4

**Table 9 foods-14-00603-t009:** Demographics and economic factors of survey participants.

List	Chinese Consumers		Thai Consumers	
	Total/Average	Percentage	Total/ Average	Percentage
Number of samples	665	100	665	100
Occupation				
Government officer	49	7.4	143	21.5
State enterprise	46	6.9	103	15.5
Company employee	256	38.5	87	13.1
Personal business	145	21.8	125	18.8
Freelance	81	12.2	84	12.6
Housewife	22	3.3	30	4.5
Students	5	0.8	56	8.4
Retired	61	9.2	37	5.6
Other	0	0	0	0
Total household income per month				
Less than RMB 4000	5	0.8	70	10.5
RMB 4001–8000	87	13.1	232	34.9
RMB 8001–12,000	342	51.4	295	44.4
More than RMB 12,000	231	34.7	68	10.2

**Table 10 foods-14-00603-t010:** Influence of sensory attributes on WTP for Chinese and Thai consumers.

Variable	Chinese Consumers				Thai Consumers			
	Coefficient	Std. Error	z	*p* > |z|	Coefficient	Std. Error	z	*p* > |z|
Taste	0.8008	0.0720	11.12	0.000	0.1569946	0.1471817	1.07	0.286
Fragrance	0.6277	0.0621	10.10	0.000	−1.177224	0.1636899	−7.19	0.000
Softness	0.4384	0.0617	7.10	0.000	0.9835731	0.1430975	6.87	0.000
Completeness	0.8168	0.0635	12.85	0.000	0.4039755	0.1370604	2.95	0.003
Standard deviation								
Taste	0.7095	0.1136	6.24	0.000	−0.3323462	0.3722423	−0.89	0.372
Fragrance	0.0581	0.1110	0.52	0.601	0.1350497	0.4472435	0.30	0.763
Softness	0.0931	0.1454	0.64	0.522	−0.3478726	0.3856231	−0.90	0.367
Completeness	−0.3548	0.1490	−2.38	0.017	−0.0105127	0.1770447	−0.06	0.953

Note: This analysis was conducted with a combined sample size of 16,625, incorporating data from both Chinese and Thai consumers to highlight differences in their preferences for fragrant rice. The log likelihood value of −3781.6969 demonstrates the model’s capacity to explain the observed data effectively. The Likelihood Ratio chi-square test value of 248.07, with 10 degrees of freedom, confirms that the model significantly improves the fit compared to a null model. The *p*-value of 0.0000, which is less than 0.001, validates the significance of the predictors in the model. This analysis underscores that sensory perception and country-of-origin factors play a critical role in shaping the purchasing decisions of both Chinese and Thai consumers, reflecting the model’s relevance in cross-national contexts.

**Table 11 foods-14-00603-t011:** Coefficients and significance of COO effects.

Variable	Chinese Consumers				Thai Consumers			
	Coefficient	Std. Error	z	*p* > |z|	Coefficient	Std. Error	z	*p* > |z|
COO—China	1.1788	0.2189	5.38	0.000	1.1788	0.2189	5.38	0.000
COO—Cambodia	1.4164	0.2124	6.67	0.000	1.4164	0.2124	6.67	0.000
COO—Vietnam	1.1936	0.2166	5.51	0.000	1.1936	0.2166	5.51	0.000
COO—Thailand	2.2779	0.2056	11.08	0.000	2.2779	0.2056	11.08	0.000
Standard deviation								
COO—China	0.6335767	0.476723	1.33	0.184	1.66425	0.3036383	5.48	0.000
COO—Cambodia	3.014759	0.4237541	7.11	0.000	0.8157004	0.2685153	3.04	0.002
COO—Vietnam	1.799721	0.4285062	4.20	0.000	0.0533708	0.2031896	0.26	0.793
COO—Thailand	7.545965	0.4831092	15.62	0.000	−1.815996	0.2604622	−6.97	0.000

Note: This analysis was conducted with a combined sample size of 16,625, incorporating data from both Chinese and Thai consumers to highlight differences in their preferences for fragrant rice. The log likelihood value of −3781.6969 demonstrates the model’s capacity to explain the observed data effectively. The Likelihood Ratio chi-square test value of 248.07, with 10 degrees of freedom, confirms that the model significantly improves the fit compared to a null model. The *p*-value of 0.0000, which is less than 0.001, validates the significance of the predictors in the model. This analysis underscores that sensory perception and country-of-origin factors play a critical role in shaping the purchasing decisions of both Chinese and Thai consumers, reflecting the model’s relevance in cross-national contexts.

**Table 12 foods-14-00603-t012:** Comparative WTP values for attributes across consumer groups.

Attributes (Variables)	Willingness to Pay Chinese Consumers	Willingness to Pay Thai Consumers
Taste	49.73	1.53
Fragrance	38.98	−11.48
Softness	27.22	9.59
Completeness	50.73	3.94
Quality standard seal	50.82	6.25
Country of origin—China	73.21	6.17
Country of origin—Cambodia	87.97	29.40
Country of origin—Vietnam	74.13	17.55
Country of origin—Thailand	141.48	73.61

## Data Availability

The original contributions presented in this study are included in the article. Further inquiries can be directed to the corresponding authors.

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
