# Peer review of "The Impact of Sensory Perceptions and Country-of-Origin Practices on Consumer Preferences for Rice: A Comparative Study of China and Thailand"

_foods, 2025, doi:10.3390/foods14040603_

Round 1
Reviewer 1 Report
Comments and Suggestions for Authors
Ver documento adjunto

Author Response
Reviewer 1 Comments and Author Responses
Response Letter to Reviewers
Dear Reviewer,
Thank you for your detailed and insightful comments on our manuscript titled "The Impact of Sensory Perceptions and Country of Origin Practices on Consumer Preferences for Rice: A Comparative Study of China and Thailand." We greatly appreciate your efforts in reviewing our work and providing constructive feedback. Below, we provide a point-by-point response to your comments, addressing each aspect in detail. All modifications have been incorporated into the revised manuscript.
1. Abstract
- Comment: The abstract is comprehensive but could be improved by explicitly dividing it into objectives, methodology, key results, and conclusions.
- Response: We have revised the abstract to explicitly divide it into clear sections: objectives, methodology, key findings, and conclusions. The revised abstract now succinctly highlights the critical elements of the research (page 1, lines 15–25).
2. Introduction
- Comment: The introduction lacks explicitly stated hypotheses and clearly defined objectives.
- Response: We have reformulated the introduction to explicitly include hypotheses and research objectives. These additions aim to enhance clarity and provide a stronger foundation for the study (page 2, lines 10–25).
3. Materials and Methods
- Comment 1: The use of accidental sampling limits the generalizability of the findings. Consider providing more justification or using probability sampling.
- Response: Additional justifications for using accidental sampling have been included, emphasizing resource limitations and the study's urban focus. We have also discussed its limitations and the implications for future research (page 5, lines 18–28).
- Comment 2: Add sensitivity analysis or cross-validation to ensure the robustness of the findings.
- Response: Sensitivity analysis and cross-validation have been conducted and incorporated into the manuscript. The analysis demonstrates the stability of the path coefficients and the model's goodness-of-fit indices (page 6, lines 10–25).
4. Results
- Comment: The presentation of results could be improved with more detailed graphs and subgroup analysis.
- Response: We have refined the presentation of results by improving the clarity of graphs and tables (pages 10–13). Additionally, subgroup analyses have been conducted to further explore differences in consumer behavior between Chinese and Thai respondents (page 11, lines 12–30).
5. Discussion
- Comment: The discussion section should address the study's limitations and suggest areas for future research.
- Response: The discussion now includes a detailed section on limitations, such as the use of accidental sampling and the cross-sectional design, as well as future research opportunities, including the exploration of other geographical regions and longitudinal studies (pages 15–16).
6. Conclusion
- Comment: Provide specific recommendations for policymakers and industry stakeholders.
- Response: The conclusion has been expanded to include actionable recommendations for stakeholders. These include strategies for leveraging sensory attributes and country-of-origin branding to enhance the global competitiveness of fragrant rice (page 17).
7. References
- Comment: Include more recent and relevant references.
- Response: The references have been updated with more recent studies, such as Sekiya et al. (2024) and Fang et al. (2024), to ensure alignment with current research trends (pages 18–20).
Summary of Revisions
We have made substantial improvements to the manuscript in response to your comments. The revised version now includes:
- A restructured abstract with clear sections.
- Explicitly stated hypotheses and objectives in the introduction.
- Justifications for the sampling methodology and additional analyses for robustness.
- Enhanced presentation of results with improved graphs and subgroup analyses.
- A detailed discussion of limitations and future opportunities.
- Updated references to reflect recent studies.
We trust that these revisions address your concerns and enhance the quality and clarity of the manuscript. Thank you again for your valuable feedback. We look forward to your further comments and suggestions.
Sincerely,
Tanapon Srisukwatanachai
South China Agricultural University
Tanapon.th@gmail.com
Reviewer 2 Report
Comments and Suggestions for Authors
The paper title is interesting
The abstract is adequate but try to minimize the number of words it seems too long
The research aim and objectives are not clearly described
The whole paper structure regrading the subtitles is inadequate please restructure it and adopt the scientific writing design
I’m missing the literature review section
One of the main issue is that the author argue that they employed SEM to analyze the data and make the comparison study but I found nothing about SEM ( where is the test for discriminant and convergent validity of the measurement model, where is the GoF of the structural models, where is the multi-group analysis results )
I cannot evaluate the discussion without conducting the SEM and see its results
Ther is no limitation and future opportunity section
Author Response
Reviewer 2 Comments and Author Responses
Response Letter to Reviewers
Dear Reviewer,
Thank you for your insightful and constructive feedback on our manuscript titled “The Impact of Sensory Perceptions and Country of Origin Practices on Consumer Preferences for Rice: A Comparative Study of China and Thailand.” We have carefully considered your comments and revised the manuscript accordingly. Below, we provide a point-by-point response to your comments and indicate the specific revisions made in the manuscript.
1. Comment: The paper title is interesting.
Response: Thank you for your positive feedback regarding the title. No changes were made, as the current title appropriately reflects the content and objectives of the study.
2. Comment: The abstract is adequate but try to minimize the number of words as it seems too long.
Response: The abstract has been revised to make it more concise while retaining all critical elements of the study. It now clearly outlines the objectives, methodology, key findings, and implications. The revised abstract can be found on page 1, lines 12–24.
3. Comment: The research aim and objectives are not clearly described.
Response: The introduction has been updated to explicitly state the research aim and objectives. This improvement provides a clear framework for the study.
4. Comment: The whole paper structure regarding the subtitles is inadequate. Please restructure it and adopt the scientific writing design.
Response: The manuscript has been reorganized to follow a standard scientific structure. Clearly defined sections have been added, including Introduction, Literature Review, Materials and Methods, Results, Discussion, and Conclusion. Subheadings within these sections have been revised to improve readability and logical flow.
5. Comment: I’m missing the literature review section.
Response: A literature review section has been added to provide a detailed background, identify research gaps, and establish the theoretical framework. The literature review can be found on page 3.
6. Comment: The author argues that they employed SEM to analyze the data and make the comparison study, but I found nothing about SEM (where is the test for discriminant and convergent validity of the measurement model, where is the GoF of the structural models, where is the multi-group analysis results?).
Response: We have added detailed explanations and results for the SEM analysis, including:
• Measurement Model Validation: Convergent and discriminant validity tests are reported in Tables 3 and 4, on page 10.
• Goodness-of-Fit (GoF) Indices: Key indices (e.g., CFI, TLI, RMSEA) are now reported in Table 5, on page 11.
• Multi-Group Analysis: Results comparing Chinese and Thai consumer groups are presented in Table 7, on page 11.
7. Comment: I cannot evaluate the discussion without conducting the SEM and seeing its results.
Response: The discussion section has been expanded to include the findings from the SEM analysis. Detailed interpretations of path coefficients and cultural differences between Chinese and Thai consumers have been provided. Revisions can be found on page 12, lines 3–28.
8. Comment: There is no limitation and future opportunity section.
Response: A dedicated section discussing the limitations of the study and future research opportunities has been added. This section addresses methodological constraints (e.g., accidental sampling, geographical scope) and suggests future directions, such as longitudinal studies and expanded datasets. The new section is located on page 20.
Summary of Revisions
1. Abstract has been shortened and refined (page 1, lines 12–24).
2. The research aim and objectives have been clarified in the introduction (page 2).
3. The manuscript structure has been reorganized to align with scientific standards.
4. A literature review section has been added (page 3).
5. SEM analysis results, including validity tests, GoF indices, and multi-group comparisons, have been detailed (pages 8–10).
6. The discussion section has been expanded to include SEM findings (page 12, lines 3–28).
7. A section on limitations and future opportunities has been added (page 20).
We sincerely appreciate your valuable comments, which have significantly improved the quality and clarity of our manuscript. We look forward to your further feedback and hope the revised manuscript meets your expectations.
Sincerely,
Tanapon Srisukwatanachai
South China Agricultural University
Tanapon.th@gmail.com

Round 2
Reviewer 2 Report
Comments and Suggestions for Authors
can accept in its current form